# Salt Tolerance of *Limonium gmelinii* subsp. *hungaricum* as a Potential Ornamental Plant for Secondary Salinized Soils

**DOI:** 10.3390/plants12091807

**Published:** 2023-04-28

**Authors:** Péter Honfi, Eman Abdelhakim Eisa, Andrea Tilly-Mándy, Ildikó Kohut, Károly Ecseri, István Dániel Mosonyi

**Affiliations:** 1Department of Floriculture and Dendrology, The Hungarian University of Agriculture and Life Science (MATE), 1118 Budapest, Hungary; 2Faculty of Horticulture and Rural Development, Department of Horticulture, John von Neumann University of Kecskemét, 6000 Kecskemét, Hungary

**Keywords:** *Limonium*, saline, salt stress, proline, NaCl, chlorophyll

## Abstract

Secondary salinization caused by climate change is a growing global problem. Searching for plants that can survive in areas with high salt content and even have decorative value was the focus of our research. Thirty plants of *Limonium gmelinii* subsp. *hungaricum* were planted in clear river sand; another thirty plants were planted in Pindstrup, a growing substrate enriched with 40% clay. With the latter, we modeled the natural soil. In addition to the control tap-water treatment, plants received 50, 125, 250, 375, and 500 mM NaCl solution irrigation twice a week. The leaf sizes of plants planted in sand decreased proportionally with the increasing NaCl concentration, and their dry matter content increased. In the clay-containing medium, leaf sizes increased, even at a concentration of 375 mM, although the dry matter content increased only at high concentrations. Carotene content in both media became higher, due to the higher NaCl concentrations, while proline content in the plants grown in sandy media increased, even with the 125 mM concentration. With our present experiment we proved the salt tolerance of the taxon, and even the soil’s great importance in supporting the plant’s salt tolerance.

## 1. Introduction

Land salinization is a global problem that affects a significant portion of the world’s irrigated farmland. According to estimates, the globe loses at least 0.5–1% of arable land each year due to salinization [1], and by 2050 half of the world’s arable land may be gone if current trends continue [2]. The primary symptoms of salt toxicity in plants include ion damage, osmotic stress, and oxidative stress [1]. These pressures negatively impact plant growth and development. Salinization can also cause physiological drought, which occurs when a plant’s roots respond to environmental stimuli such as a lack of water or an abundance of salt ions in the soil [3]. This lack of water has detrimental impacts on plants, including the accumulation of reactive oxygen species and oxidative stress, and reduced photosynthetic rates [3]. Plant cells have sodium receptors that detect increases in sodium, which elevates cytosolic calcium and triggers adaptation responses [4]. Through the SOS pathway, the extra cytosolic sodium is either sequestered in vacuoles or evacuated from the cell [5,6]. Growing traditional crops in areas with high salt content is uncertain, as most plants cannot survive in such environments. Using molecular biology and genetic engineering to create salt-tolerant plants is a possibility, but it can be costly and time-consuming. An alternative solution is to cultivate naturally salt-tolerant plants in these areas, as it is more affordable, sustainable and cost-effective [7].

Halophytes are able to thrive and carry out their entire life cycle in regions where the concentration of NaCl is above 200 mM [8] and could be optimized to minimize the salinity of salty soils [9]. *Limonium* is a biodiverse halophyte taxon within the *Plumbaginaceae* family, known for its remarkable characteristics, and has the ability to thrive in saline environments. This taxon comprises around 370 species with high adaptability to such conditions [10]. Continental climates are ideal for the growth of *Limonium* species [11]. Nearly all species of *Limonium* are endemic to salty environments, and several of them can thrive in extremely salty conditions [12]. *Limonium gmelinii* subsp. *hungaricum*, also known as Hungarian sea lavender or Hungarian statice, is a perennial herbaceous plant of the genus. The species is indigenous to the coastal areas of Eastern Europe and Asia and is known for its tolerance to salt and drought. It typically grows to a height of 30–60 cm, and has a woody base with branching flower stems. The leaves are linear-lanceolate, sessile, and fleshy, and range in size from 15 to 30 cm long and 5 to10 cm wide. The flowers are small, pinkish-purple and arranged in dense spikes. They bloom in late summer and early fall [13]. The plant has developed a tolerance for high levels of salt in the soil, which allows it to thrive in environments such as coastal dunes, salt marshes, and sandy soils, making it a potential ornamental plant for secondary salinized soils [14].

Halophytic plants, which possess the ability to thrive in environments with high salinity, exhibit various modifications in terms of morphology, physiology, biochemistry, and molecular structure [15]. The adaptation mechanisms of halophytic plants in response to salinity stress include: preservation of the photosynthetic system through enhanced chlorophyll synthesis [16], adjustment of carotenoid levels [15], modulation of reactive oxygen species (ROS) levels, stimulation of enzymatic antioxidant activity [17], and increased production of non-enzymatic antioxidants such as proline [18].

*Limonium* sp. or recretohalophytes are distinctive halophytes that possess specialized structures on their epidermis, such as salt bladders and salt glands, which allow them to cope with salt-rich environments [19]. Recretohalophytes secrete excess salt ions through their salt glands in order to maintain ionic balance [20,21]. The studies by Daraban et al. and Rozentsvet et al. [22,23] have shown that the salt glands play a crucial role in the adaptation of the czrynohalophyte *L. gmelinii* (Willd.) to high-salt conditions; crynohalophytes are plants that secrete excess salt on their leaves’ surface [24]. According to the findings of Leng et al. [25] *L. gmelinii*’s salt glands show a distribution pattern that suggests an optimal concentration of 50 mM NaCl for its growth. In contrast, a study by [21] showed that the salinity threshold of *L. gmelinii* can reach up to 400 mM. Under environmental stress conditions, plants typically exhibit a reduction in growth, as they prioritize activating their defense mechanisms using their metabolic precursors and energy resources rather than allocating them towards biomass accumulation, according to Munns and Tester [1]. An investigation by Ghanem et al. [26] discovered that the buildup of proline was connected to a decrease in biomass, chlorophyll a/b ratio, and carotenoid levels in *Salicornia europaea* when exposed to high salinity. According to MI et al. [21], *Limonium* species subjected to saline stress exhibited a positive correlation between their total fresh weight and variables such as total dry weight, chlorophyll content, and intercellular CO_2_ concentration. Additionally, many *Limonium* species have been found to contain proline or osmolytes, which are important in preserving osmotic equilibrium and safeguarding plants against stress, especially at elevated levels of NaCl, serving as protectors of plasma membrane integrity, ROS scavengers, or signaling molecules during severe stress [8,27].

Additionally, salinity can be controlled through the implementation of appropriate soil amendments and effective water management techniques [28]. The physical characteristics of the soil, such as texture and structure, can also impact the growth and performance of halophytic plants, as they affect the soil’s nutrient retention, ventilation, water retention and drainage capacity [29]. In this study, we aim to examine the response of *Limonium gmelinii* subsp. *hungaricum* to saline conditions and the influence of soil type on its salt tolerance.

## 2. Results

### 2.1. Effect of NaCl Stress on the Growth and Development of Limonium gmelinii subsp. hungaricum in Sandy and Clayey Soil

The results of *Limonium gmelinii* subsp. *hungaricum* grown in sandy and clayey soil were analyzed to determine the effect of NaCl treatments on the plant’s growth and development. The treatments were given in various concentrations of 50 mM, 125 mM, 250 mM, 375 mM, and 500 mM. In addition, the control treatment was given in the form of normal irrigation or tap-water irrigation, which represented the plant’s growth without stress (Figure 1).

In the sandy soil, the fresh weight of *Limonium gmelinii* subsp. *hungaricum* showed a decrease of about 17.4% to 57% compared to the control treatment. Similarly, the dry weight showed a decrease of about 10.1% to 39.3% compared to the control treatment (Figure 2A), with NaCl concentrations ranging from 50 mM to 500 mM. On the other hand, the relative dry weight (g/100 g) in sandy soil increased by about 9% to 41% compared to the control treatment with NaCl concentrations, ranging from 50 mM to 500 mM (Figure 2B). The length of the leaves was also significantly affected by the NaCl treatments, with a decrease of approximately 4% to 31% compared to the control treatment, with NaCl concentrations ranging from 50 mM to 500 mM. The width of the leaves showed a similar pattern, with a decrease of approximately 6% to 29% compared to the control treatment, with NaCl concentrations ranging from 50 mM to 500 mM (Figure 2C).

In the case of clayey soil, the results showed that the relative dry weight increased with the increase in the NaCl concentration, with the highest value at 500 mM (43 g/100 g) (Figure 2E). However, the fresh weight showed an increasing trend until 250 mM, where it showed an increase of 12% compared to the control and then decreased at 375 mM by 23% and at 500 mM by 10% compared to the control. The same trend was observed in the dry weight, with an increase until 375 mM, where it showed an increase of 12% compared to the control. However, at 500 mM NaCl treatment, the dry weight decreased by 19% compared to 375 mM treatment and by 9% compared to the control (Figure 2D). This suggests that the plants in the clayey soil were able to withstand the salt stress to some extent but showed reduced growth and biomass production as the salt concentration increased beyond 375 mM.

The leaf length and width of plants grown in the clayey soil showed varying responses to the NaCl treatments (Figure 2F). The leaf length initially increased with increasing NaCl concentration, reaching its maximum at 375 mM by 17% compared to the control. However, the leaf length decreased significantly at 500 mM NaCl concentration, by 4% compared to the control. On the other hand, the leaf width gradually increased with increasing NaCl concentrations up until 375 mM, compared to the control (Figure 2F). However, the leaf width decreased by 3% with a 500 mM NaCl treatment, indicating that there is an optimal range of NaCl concentrations for *Limonium gmelinii* subsp. *hungaricum* to grow optimally. These observations suggest that NaCl stress has a differential effect on the different morphological characteristics of plants grown in clayey soil.

### 2.2. The Effects of NaCl Concentration on Chlorophyll and Carotenoid Content in Limonium gmelinii subsp. hungaricum Grown in Sandy and Clayey Soils

The pigment content of *Limonium gmelinii* subsp. *hungaricum* was subjected to varying levels of NaCl in both sandy and clay soils. In sandy soil (Figure 3A), the results indicated that the control group exhibited a relatively constant level of chlorophyll, with a modest reduction observed at 50 mM NaCl, followed by an increase at 125 mM NaCl. The highest chlorophyll concentration was recorded at 375 mM NaCl, demonstrating a 14.4% increase compared to the control group, before experiencing a slight decrease at 500 mM NaCl, and a slight increase at 250 mM NaCl. On the other hand, the carotenoid content in sandy soil (Figure 3B) exhibited a divergent trend, with a modest reduction observed at 50 mM NaCl, followed by a slight uptick in carotenoid concentration at 125 mM NaCl, and then a significant increase observed with higher NaCl concentrations, peaking at 375 mM NaCl with an increase of 16.4% compared to the control group. This increase was further amplified at 500 mM NaCl, where the concentration of carotenoids reached an 18.6% increase compared to the control group.

In clayey soil, the results indicated that the control group had a moderate concentration of chlorophyll (Figure 3C), which was diminished at 50 mM NaCl. However, this reduction was followed by an increase in chlorophyll concentration at 125 mM NaCl. The highest concentration of chlorophyll was recorded at 250 mM NaCl, where the concentration increased by 38% compared to the control group, before experiencing a decrease at 375 mM NaCl, and a slight increase at 500 mM NaCl. Similarly, the carotenoid content in clayey soil (Figure 3D) showed a decrease at 50 mM NaCl, followed by an increase at 125 mM NaCl. The highest concentration of carotenoids was observed at 500 mM NaCl, where the concentration increased by 70% compared to the control group, followed by a stable level at 375 mM NaCl and 250 mM NaCl. 

### 2.3. Effect of NaCl Levels on Sodium Content in Limonium gmelinii subsp. hungaricum Leaves

The data in Figure 4 provide valuable information on the effect of increasing NaCl levels on the sodium (Na) content in the leaves of *Limonium gmelinii* subsp. *hungaricum* grown in sandy substrate. The control treatment recorded an average Na content of 1.13 g/100 g fresh weight. With a steady increase in NaCl levels, the Na content in the leaves also increased. At the highest NaCl level of 500 mM, the average Na content reached 3.47 g/100 g fresh weight, representing a 205% increase on the control treatment. This result demonstrates the remarkable ability of *Limonium gmelinii* subsp. *hungaricum* to regulate its internal Na levels in response to changing external conditions.

### 2.4. Proline Content Response of Limonium gmelinii subsp. hungaricum Leaves to Increasing NaCl Concentrations in Sandy and Clayey Soils

In sandy soil (Figure 5A), the proline content of *Limonium gmelinii* subsp. *hungaricum* leaves was observed to increase significantly with increasing NaCl concentrations. The control group, with no added NaCl, had a proline content of 0.128 µmol g^−1^ FW. However, when exposed to 50 mM of NaCl, the proline content increased by over 80%. This trend continued with even higher concentrations of NaCl, with the proline content reaching a peak of over 1100% greater than the control group at 500 mM. Interestingly, at 375 mM, the proline content dropped slightly, but still remained well above the control group.

In clayey soil (Figure 5B), the proline content also increased with increasing NaCl concentrations, but to a much lesser extent than in sandy soil. The control group in clayey soil had a proline content of only 0.005 µmol g^−1^ FW. Even at the highest concentration of 500 mM, the proline content only increased by over 200%. The proline content in clayey soil was consistently lower than that of sandy soil at each NaCl concentration. Overall, it can be inferred that *Limonium gmelinii* subsp. *hungaricum* leaves are better adapted to handling salt stress in sandy soil than in clayey soil. 

## 3. Discussion

Salt stress induced by NaCl is one of the most frequent and widely studied abiotic stresses in controlled and natural environments [30]. Plants are subjected to high-salt conditions, which might impair photosynthesis, and diminish the fresh weight and dry matter accumulation in the plant [21]. The detrimental effects on the formation of plant biomass are attributed to osmotic stress, nutritional insufficiency, and ionic toxicity induced by NaCl, which impacts physiological status [31,32]. This is compatible with findings that NaCl inhibits the plant’s capacity to absorb water, resulting in delayed growth; additionally, when there is an excessive amount of salt present in the transpiration stream, it can eventually lead to damage to the cells in the leaf that is undergoing transpiration, thereby further inhibiting growth [1].

Understanding how halophytes respond to salt is crucial to their further evolution and practical application. It is possible to extrapolate a plant’s salt tolerance from its threshold of salt tolerance. In this investigation, we examined to what extent *L. gmelinii* subsp. *hungaricum* can tolerate salt stress and the vital role of soil texture in ameliorating the negative impact of saline conditions.

According to the findings illustrated in Figure 2, NaCl treatments had a significant impact on the growth and development of *Limonium gmelinii* subsp. *hungaricum* in both sandy and clayey soil. In the sandy soil (Figure 2A–C), the fresh weight and dry weight of the plants decreased with increasing NaCl concentration, ranging from 17.4% to 57% and 10.1% to 39.3%, respectively, compared to the control treatment. This decrease in fresh weight and dry weight was accompanied by a decrease in the length and width of the leaves. Likewise, Mi et al. noticed a significant reduction in the fresh weight of all four Limonium species, namely *L. aureum*, *L. gmelinii (Willd.) Kuntze*, *L. otolepis*, and *L. sinuatum* Mill., when exposed to a concentration of 400 mM NaCl. Among these species, *L. gmelinii* showed the least reduction in leaf fresh weight, with only 49% of that of the control [21]. Low or moderate salinity stimulates growth in a few salt-tolerant halophytes, whereas salt above a species-specific concentration threshold inhibits growth [33]; hence, the growth parameter is a vital indicator for assessing the degree of salt tolerance of the plant [34]. Likewise, Jangra et al. [35] observed that salinity reduced the fresh weight, dry weight, and leaf area of *Sorghum* plants. In a comprehensive study by González-Orenga et al. [14] on different *Limonium* species (*L. albuferae*; *L. dufourii*; *L. girardianum*; *L. narbonense*; *L. santapolense*; *L. virgatum*), it was investigated that growth was enhanced at 200 and even 400 mM NaCl, and was only suppressed at higher salt levels. Mi et al. [21] determined the salinity-tolerated threshold of *L. gmelinii* at 420 mM NaCl, while our work reached 500 mM in most estimated parameters. Therefore, our results suggest that the optimal growth of *L. gmelinii* is under low NaCl levels, and it could maintain their survival under moderate and high NaCl levels (250 and 500 mM). 

On the other hand, in the clayey soil (Figure 2D–F), the relative dry weight increased with increasing NaCl concentration up until 375 mM, where it showed an increase of 12% compared to the control treatment. Beyond 375 mM, the dry weight decreased by 19%. The leaf length and width showed varying responses to the NaCl treatments in the clayey soil, with the leaf length reaching its maximum at 375 mM and the leaf width gradually increasing up until 375 mM. However, beyond 375 mM, both the leaf length and width decreased, suggesting that there is an optimal range of NaCl concentrations for *Limonium gmelinii* subsp. *hungaricum* to grow optimally. The different effects of NaCl treatments on the growth and development of *Limonium gmelinii* subsp. *hungaricum* in sandy and clayey soil are likely due to differences in soil properties, such as water-holding capacity, nutrient availability, and ion toxicity. Similarly, [36], the absence of plant biomass losses in a clayey medium implies that finer textures mitigate salinity stress compared to a coarser soil texture (sandy soil).

Photosynthesis is a useful physiological metric for assessing plants’ response to salinity stress and determining stress-tolerant species, as it directly represents the plants’ photosynthetic capacity. Salinity stress is a known inhibitor of plant photosynthesis [37], and, consequently, biomass accumulation [21]. As non-enzymatic antioxidants, carotenoids are essential in preserving the photosynthetic system and regulating plant growth, in which plants synthesize abscisic acid from carotenoids via the mevalonic acid pathway under stressful conditions [38]. Our results in both soil types suggests that the salt tolerance of *L. gmelinii* evaluated in this work is primarily attributable to variations in the degree of photosynthetic pigment damage caused by increasing salt concentration (Figure 3A–D). In sandy soil, the results showed that the control group had a constant level of chlorophyll, with a modest reduction observed at 50 mM NaCl and then an increase at 125 mM NaCl. The highest concentration of chlorophyll was recorded at 375 mM NaCl, showing a 14.4% increase compared to the control group. On the other hand, the carotenoid content in sandy soil exhibited a divergent trend, with an increase observed with higher NaCl concentrations, peaking at 375 mM NaCl with an 18.6% increase compared to the control group.

In clayey soil, the results showed that the control group had a moderate concentration of chlorophyll, which decreased at 50 mM NaCl, but then increased at 125 mM NaCl. The highest concentration of chlorophyll was recorded at 250 mM NaCl, where the concentration increased by 38% compared to the control group. The carotenoid content in clayey soil showed a similar trend, with the highest concentration observed at 500 mM NaCl, where the concentration increased by 70% compared to the control group. This is further supported by the findings of [39], who reported that halophytes utilize specific mechanisms to protect themselves from photo-damage, such as dissipating excess energy generated during photosynthesis. Additionally, [40] found that the levels of Chl a, Chl b, total chlorophyll, Chl a/b, and carotenoids in *S. maritima* seedlings were unchanged in response to salinity stress, further supporting the importance of these pigments in halophyte tolerance to salt. A comparable study confirmed a correlation between the increase in carotenoids in two types of halophytes, *Arthrocnemum macrostachyum* and *Sarcocornia fruticose*, and enhanced salt tolerance, as a method to maintain chlorophyll amount, along with different NaCl levels [26].

In halophytes species, the salt accumulation of the plants’ aerial part sections matched their salt-tolerant strategy [22]. By secreting extra sodium and potassium salts, *L. gmelinii* plants can adjust their salt level [23]. The findings from the study of *Limonium gmelinii* subsp. *hungaricum* grown in sandy substrate reveal the plant’s capability of regulating its internal sodium (Na) content in response to external conditions. As observed in Figure 4, the control treatment recorded an average Na content of 1.13 g/100 g fresh weight. With the progressive increase in NaCl levels, the Na content in the leaves also increased, reaching a maximum of 3.47 g/100 g fresh weight at 500 mM NaCl, a 205% increase on the control treatment. This result is in agreement with previous studies that have demonstrated the ability of plants to respond to salinity stress by adjusting their internal Na levels. For instance, a study by [25] showed that increasing NaCl concentrations resulted in a significant increase in Na content in the leaves of halophytes, including *Limonium bicolor* and *Limonium gmelinii*. Several studies have explored the underlying mechanisms of Na uptake and transport in plants. For example, a study by [41] found that the expression of genes involved in Na uptake and transport was regulated in response to increasing NaCl levels in the leaves of the halophyte *Suaeda salsa*. Additionally, halophytes, specifically recretohalophytes, have a unique advantage over other halophytes and non-halophytes in maintaining cellular ion homeostasis. They are able to directly secrete high levels of salt ions onto the leaf surface through specialized structures called salt glands. This makes them exceptional in handling the challenges of salinity compared to other classes of plants [22,42]. Salty glands regulate osmotic pressure and promote salt tolerance by preserving ion equilibrium [43]. Sodium glands regulate the leaf’s internal ionic composition in the chloroplast and cytosol, which prevents leaf cell dehydration and enhances photosynthesis [14]. 

Proline is a non-essential amino acid that can adjusts osmotic pressure and act as an osmoprotectant, helping the plant to maintain its water balance in the face of high salt concentrations [44]. It is also a powerful antioxidant, helping the plant to protect itself against the damage caused by high salt concentrations [27]. In halophytes, osmotic equilibrium is accomplished by accumulating uncostly energy ions, such as Na^+^ and Cl^−^, and osmolyte solutes with low molecular weight [45]. 

In sandy soil, the leaves were able to produce a significantly higher amount of proline, a molecule known to help protect plants from the damaging effects of salt (Figure 5A). As NaCl concentrations increased, the proline content in the leaves of these plants skyrocketed, reaching levels over 1100% higher than the control group, in the most extreme condition. In contrast, the proline content in clayey soil was much lower, and despite increasing with higher NaCl concentrations, it never reached the same levels as in sandy soil (Figure 5B). This suggests that *Limonium gmelinii* subsp. *hungaricum* leaves are adapted to handle salt stress in sandy soil. Similar results have been found in numerous halophyte species, where proline content increased in response to salt stress [46]. Additionally, the substantial accumulation of proline observed at a high NaCl level in irrigation water compared to unstressed conditions and the observed fluctuations in proline content may be caused by subsequent mitochondrial damage and repair during the pro-oxidation process [47,48]. *Limonium* species’ tolerance to salinity was mainly based on the active transport and buildup of ions in the leaves, along with the simultaneous production of soluble sugars and proline as compatible solutes for osmotic regulation. The halophyte cytoplasm’s synthesis and the accumulation of osmolytes maintain osmotic equilibrium under stress, to compensate for inorganic ion accumulation in vacuoles [14].

The results from the study of *Limonium gmelinii* subsp. *hungaricum* in clayey soil were quite intriguing. The plants showed a slight improvement in growth and development, even when subjected to increasing salt concentrations (Figure 2D–F), and the lower proline content with increasing NaCl level until 375 mM compared to that in sandy soil (Figure 5B), despite the same concentrations of NaCl being applied. This may indicate that soil structure is closely connected to the soil’s ability to adsorb or desorb chemical ions [49], so a portion of the Na^+^ in the salty water was likely absorbed and accumulated by clayey soil particles and became less accessible to the plant, due to the clayey soils’ lower leaching rate and larger soil surface area, which can help to reduce ion toxicity and improve plant growth [50]. As a result of their role as sites of cation exchange for soils, clays also serve as a source of extra exchangeable Na^+^ [49]. Thus, clayey soil with finer texture retains more water, reducing drought-related salinity stress and production losses, as the ionic components of finer-textured soils are regarded as macronutrients, a feature attributed to their response to specific ion impacts [36]. On the contrary, sandy soils have limited nutrient- and water-holding capacity and high infiltration rates [29]. In general, sandy soils would be preferred for irrigation at highly saline levels [51]; however, more essential plant nutrients leached from the sandy substrate than from clay-textured media during the applied irrigation [50]. Panta et al. [50] hypothesized that soil texture and nutrient-holding ability may have played a role in the increase in chlorophyll content and stomatal conductance, as well as the naturally high nutritional content of the clay soil, which may have benefited plant growth. However, beyond a certain concentration, the accumulation of ions in the soil can become toxic and result in reduced growth and development, as seen in the decrease in dry weight and leaf length and width of the plants grown in both soils at 500 mM NaCl treatment (Figure 2). 

This is the first study of *Limonium gmelinii* subsp. *hungaricum* in clayey soil and it has exhibited the ability to survive and grow better than in sandy soil, despite increasing salt concentrations. However, further research is needed to confirm these findings and to determine the mechanisms behind these changes.

## 4. Materials and Methods

### 4.1. Material, Cultivation Conditions, and Trial Layout

Seeds of *Limonium gmelinii* subsp. *hungaricum* were collected in the Apajpuszta region, Hungary, in October 2019. A total of 3–4 leaf-containing plugs were grown from these seeds in 104-hole Teku^®^ trays in Pindstrup Blond Gold substrate (Figure 6). A total of 180 plants were potted in pure river sand in 7 × 7 × 8 cm black Teku^®^ containers, in order to model the salt tolerance of the species. Another 180 plants were potted in 40% clay containing Pindstrup Blond Gold substrate in 7 × 7 × 8 cm black Teku^®^ containers, in order to model the original soil. The experiment was carried on in a glasshouse, with day temperature 20 ± 5 °C, night temperature 16 ± 5 °C, and relative air humidity 60 ± 5%. Based on the experiment of Geissler et al. (2009), plants of both groups were irrigated with different concentrations of NaCl as follows: control with tap water, and further groups with 50, 125, 250, 375 and 500 mM NaCl solution twice a week, in a 50 mL container. The treatment solution of plants grown in sand was enriched with 2 g/L Yara Mila Ferticare 14:10:18 NPK fertilizer. The experiment was set up in a random block layout, with 6 repetitions, 5 plants per repetition, and each pot containing 1 plant.

### 4.2. Morphological Characteristics

After 12 weeks, the morphological parameters of the plants were measured: leaf length, plant diameter and leaf diameter were estimated using a measuring tape. Since the plant diameter is roughly twice the leaf length, this parameter was not measured in the experiment planted in sand. The fresh weight of 10 randomly chosen plants per treatment was calculated. The root and shoot were then placed in separate paper bags and dried in an oven. The dry weight of the root and shoot was recorded after the samples were dried at 105 °C for 15 min and 75 °C for three days [52]. The dry matter content was determined by dividing the dry weight of the leaves by the fresh weight, and expressed as a percentage [53]. The height of the plants was measured using a tape measure.

### 4.3. Chlorophyll, Carotenoid and Proline Determination

Leaves of the same position were taken from 6 × 5 plants per treatment; an average sample was prepared from them and used for the following determination:

The measurement of photosynthetic pigments involved grinding 0.1 g of fresh weight (FW) of leaves in 80% cold acetone and centrifuging the mixture at 5000× *g* for 10 min. The absorbance of the purified chlorophyll samples was then recorded using a UV–VIS Spectrophotometer (Genesys 10S, Waltham, MA, USA) at 470, 646, and 663 nm. The chlorophyll and carotenoid contents were calculated based on the method described by [54].

The impact of the treatments on the stress state of the plants was assessed by determining the level of proline, using a method described in Reference [55]. This involved grinding 300 mg of fresh leaves in a solution containing 3% aqueous sulfosalicylic acid and filtering the mixture through Whatman no. 1 filter paper. The resulting filtrate was mixed with acid ninhydrin and glacial acetic acid and heated at 100 °C for 1 h, then cooled on ice for 15 min. Toluene was added, and the mixture was vortexed for 20 s before measuring the absorbance of the upper phase at 520 nm, using a spectrophotometer. The proline concentration was then calculated and reported as μmolg^−1^ FW.

### 4.4. Determination of Sodium Content in Leaves Grown in Sand

The condition of the leaf samples grown in sand was determined using a flame photometer, following the method outlined by [56]. A dry plant sample of 2 g was ground to a particle size of 2 mm and incinerated in a muffle furnace at 600 °C for 4 h. The ashes were treated with 40 mL of 30% HCl and 5 mL of HNO_3_, filtered and cooled, and the resulting solution was diluted with deionized water to a volume of 250 mL. The sodium (Na^+^) content was then determined, using the flame photometer Na^+^ content in leaves.

### 4.5. Statical Analysis

The experiment was set up in a completely randomized design. The Two-Way MANOVA followed by UNIANOVA for the variables with Bonferroni’s correction was run for all dependent variables, between factors at two levels: 1.Treatments (control, no salinity stress) and NaCl treatments (50, 125, 250, 375 and 500 mM); 2. Soil type: sandy and clayey. It was assumed that the normality of the residuals for all dependent variables were accepted by the Kolmogorov–Smirnov test (*p* > 0.05) [57]. The homogeneity of variances established by Leven’s F test was satisfied for all dependent variables *p* > 0.05 [58]. Tukey’s post hoc test was used for factor level comparisons [59,60]. Pairwise within-subject effects were compared using Bonferroni’s method. All statistics were conducted using the software IBM SPSS27, Armonk, NY, USA [60].

## 5. Conclusions

In conclusion, the effect of NaCl stress on the growth and development of *Limonium gmelinii* subsp. *hungaricum* in sandy and clayey soil was analyzed. The results showed that the plant’s growth and development were impacted by the NaCl treatments in varying ways. In the sandy soil, the plant’s fresh weight and dry weight decreased, while its relative dry weight increased. The leaf parameters of length and width were also impacted, with a decrease observed in both. The decrease in biomass coincide with increases in proline concentration and leaf NaCl content with higher salt stress. In the clayey soil, the relative dry weight increased with increasing NaCl concentration, while the fresh weight showed a peak at 250 mM NaCl and decreased thereafter. The parameters of the leaves (length and width) increased until 375 mM NaCl and decreased thereafter. Additionally, a significant proline accumulation increment was seen only with the higher NaCl levels in clayey soil. The pigment content of the plant was also impacted by the NaCl treatments, with a moderate reduction observed at 50 mM NaCl, followed by an increase at higher NaCl concentrations, peaking at 375 mM NaCl for both chlorophyll and carotenoid content in sandy soil, and 250 mM NaCl for chlorophyll content and 500 mM NaCl in clay soil. These results suggest that there is an optimal range of NaCl concentrations, ranging from 50 to 125 mM, for *Limonium gmelinii* subsp. *hungaricum* to grow optimally, while 500 mM NaCl had the most negative effect on *L. gmelinii* growth, and that substrate texture also plays a role in the plant’s response to salt stress.

## Figures and Tables

**Figure 1 plants-12-01807-f001:**
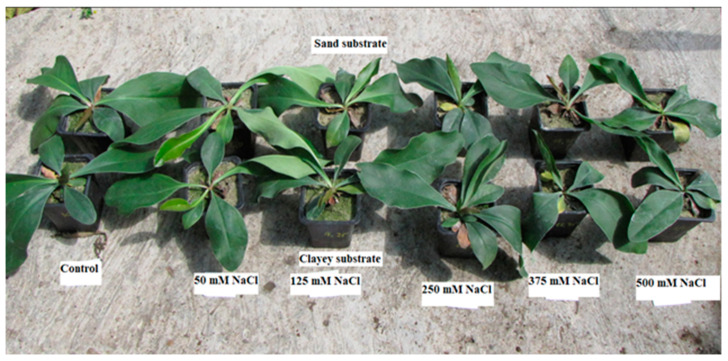
Effect of NaCl stress on morphological characters of *Limonium gmelinii* subsp. *hungaricum* under sandy soil and clayey soil.

**Figure 2 plants-12-01807-f002:**
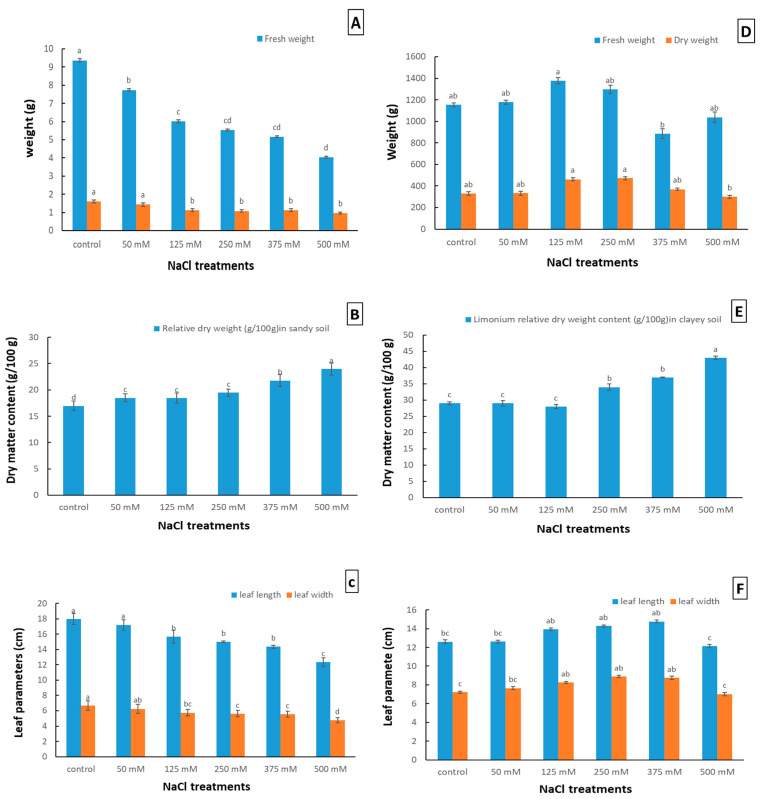
Effect of NaCl stress on morphological characters of *Limonium gmelinii* subsp. *hungaricum* under sandy soil and clayey soil. Leaves’ fresh weight and dry weight (g) in sandy soil (**A**), dry matter content (g/100 g FW) in sandy soil (**B**), and leaf parameters (cm) in sandy soil (**C**) of *Limonium gmelinii* subsp. *hungaricum* under different NaCl levels (Mm). Leaves’ fresh weight and dry weight (g) in clayey soil (**D**), dry matter content (g/100 g FW) in clayey soil (**E**), and leaf parameters (cm) in clayey soil (**F**) of *Limonium gmelinii* subsp. *hungaricum* under different NaCl levels (Mm), data represent averages of ten replications ± SD. The various letters indicate a significant variance between treated plants. *p* < 0.05 from Tukey’s multiple-range test.

**Figure 3 plants-12-01807-f003:**
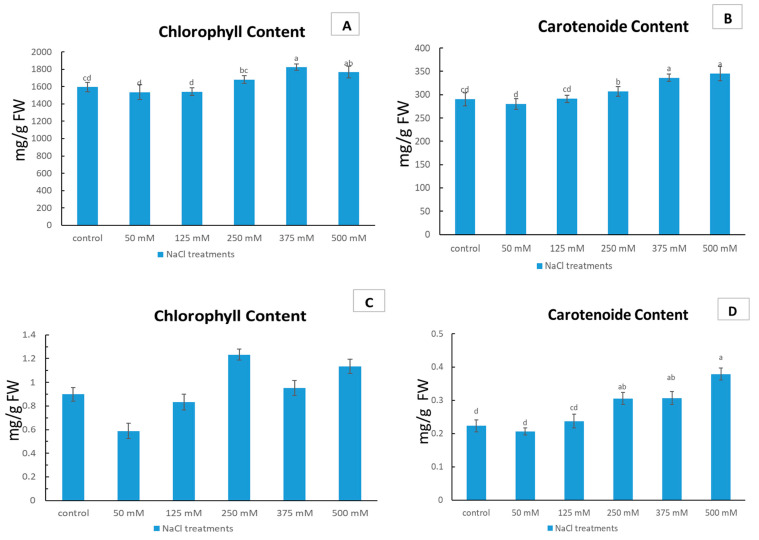
Effect of NaCl stress on *Limonium gmelinii* subsp. *hungaricum* under sandy soil and clayey soil. Total chlorophyll content mg/g FW (**A**) and carotenoid content mg/g FW (**B**) in sandy soil and total chlorophyll content mg/g FW (**C**) and carotenoid content mg/g FW (**D**) in clayey soil of *Limonium gmelinii* subsp. *hungaricum*, under different NaCl levels. Data represent averages of ten replications ± SD. The various letters indicate a significant variance between treated plants. *p* < 0.05 from Tukey’s multiple-range test.

**Figure 4 plants-12-01807-f004:**
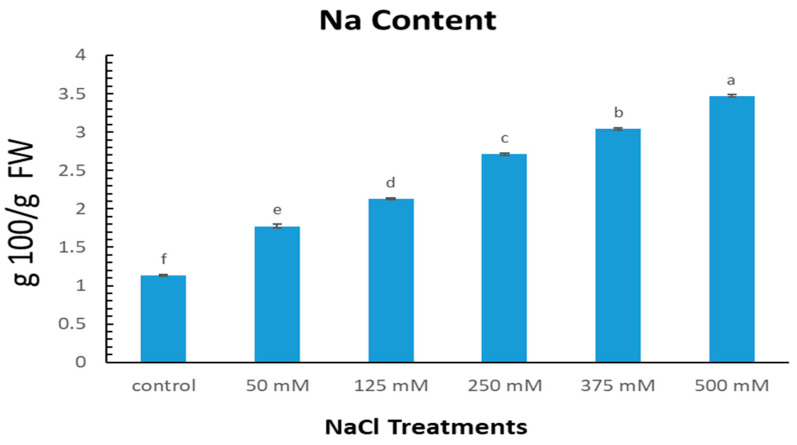
Leaves’ Na content of *Limonium gmelinii* subsp. *hungaricum* in sandy soil under different NaCl levels (mM). Data represent averages of ten replications ± SD. The various letters indicate a significant variance between treated plants. *p* < 0.05 from Tukey’s multiple-range test.

**Figure 5 plants-12-01807-f005:**
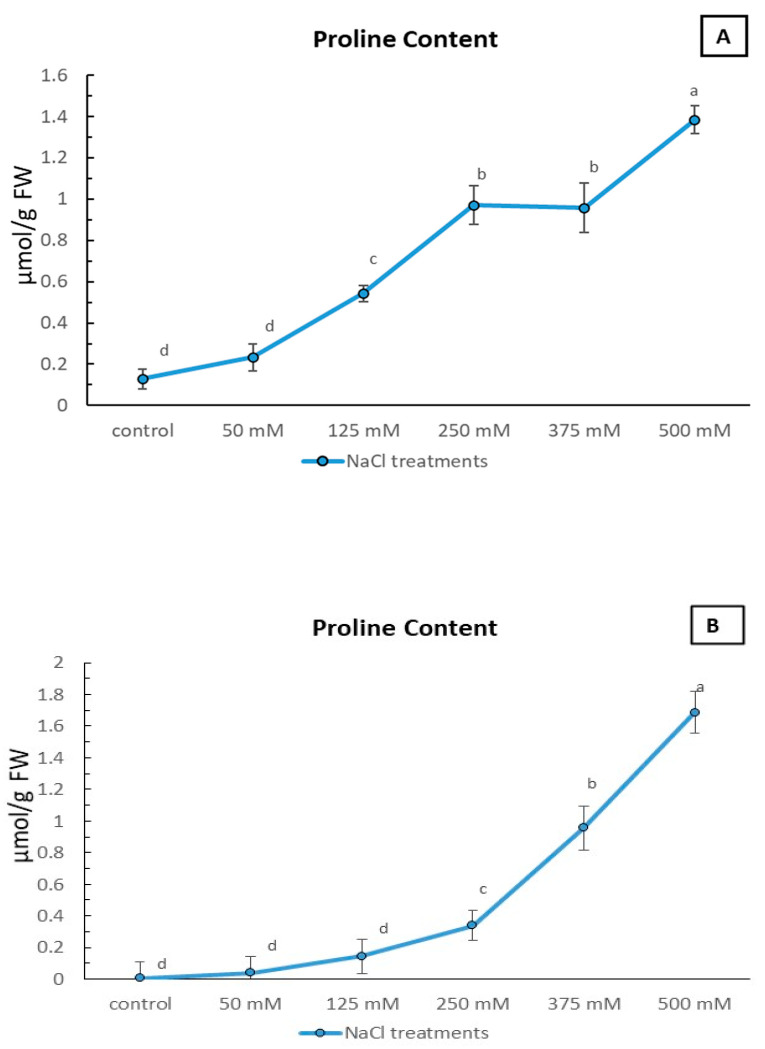
Effect of NaCl stress on proline content under sandy soil and clayey soil. Proline content of *Limonium gmelinii* subsp. *hungaricum*, under different NaCl levels in sandy soil (**A**) and clayey soil (**B**); data represent averages of ten replications ± SD. The various letters indicate a significant variance between treated plants. *p* < 0.05 from Tukey’s multiple-range test.

**Figure 6 plants-12-01807-f006:**
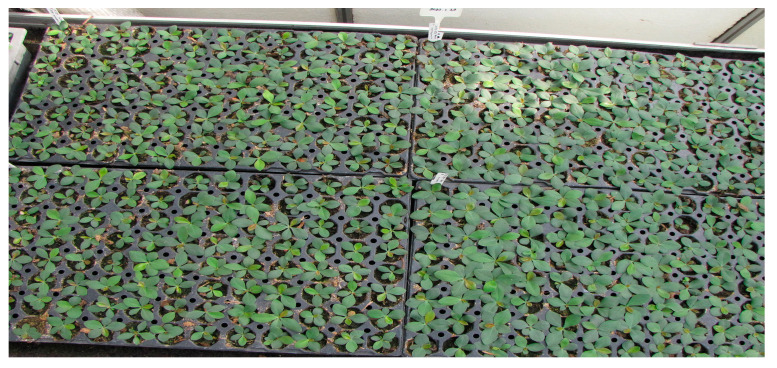
3–4 leaf-containing plugs in Teku^®^ trays in Pindstrup Blond Gold substrate.

## Data Availability

Not applicable.

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
