# Peer review of "Salt Tolerance of Limonium gmelinii subsp. hungaricum as a Potential Ornamental Plant for Secondary Salinized Soils"

_plants, 2023, doi:10.3390/plants12091807_

Round 1

Reviewer 1 Report

The objective of the paper is attractive, especially from a breeding perspective. However, the manuscript submitted to the review is not ready for being published in a scientific journal. Detail and comments are included in the attached paper.

In my opinion, the Conclusions section should have been more precise. There is no information on which particular value of the NaCl salt concentration is critical for the plants tested. What type of substrate is better for salt stress? Which soil type is better for salt stress? Was the substrate used in the experiment tested for physico-chemical properties, or do the authors base their conclusions on literature data?

Reviewer 2 Report

The manuscript of paper by Honfi et al. ‚Salt tolerance of Limonium gmelinii subsp. hungaricum as a potential ornamental plant for secondary salinized soils’ contains a comprehensive analysis of possible mechanisms of Limonium gmelinii subsp. hungaricum plant tolerance to salinization. The experiment was very well designed and the physiological methods are appropriated as well as is written in good English. The discussion is not superficial and includes a critical analysis of the results.

Firstly, lack is the schematic diagram of the experimental setup which would be useful to understand experiment design. 

Secondly, Authors showed only Na ion content in the examined plants, I do not have to explain how important is measurement other ions such as magnesium, potassium, iron, etc… and necessary to explanation to ions disturbance during salt stress.

In such a physiological work Authors should presented more pictures illustrating whole plants as well as leaves morphological changes after salt treatment.

Taking all mentioned above into account, the manuscript would merit publishing in Plants, but it requires major revision of Results. After the correction it can be recommended for publishing. 
